# Continuous and Secure Integration Framework for Smart Contracts

**DOI:** 10.3390/s23010541

**Published:** 2023-01-03

**Authors:** Alvaro Reyes, Miguel Jimeno, Ricardo Villanueva-Polanco

**Affiliations:** Department of Computer Science and Engineering, Universidad del Norte, Barranquilla 080007, Colombia

**Keywords:** DevOps, automated tools, cloud, continuous integration, continuous deployment, smart contracts

## Abstract

As part of agile methodologies seen in the past few years, IT organizations have continuously adopted new practices in their software delivery life-cycle to improve both efficiency and effectiveness of development teams. Two of these practices are continuous integration and continuous deployment, which are part of the DevOps cycle which has helped organizations build software effectively and efficiently. These practices must be considered for new technologies such as smart contracts, where security concerns and bugs might cost more once deployed than traditional software. This paper states the importance of using a proper DevOps routine and how it is possible to apply this practice to a smart contract build. Specifically, this paper introduces a framework to implement DevOps for smart contracts development by describing multiple DevOps tools and their applicability to smart contract development.

## 1. Introduction

In software development projects, organizations work to implement development practices using mostly agile methodologies [1]. One of these practices is DevOps. This software development practice focuses on improving the cooperation between the different teams working on a project, especially the development and operation team, from which the name is derived [2,3]. The result is an improvement in the productivity of all involved teams by effectively using their time, which leads to faster software development cycles and higher product quality [1,4]. However, there are issues in implementing such practices, as presented in works by [5,6], and others. Typical issues include manual and time-consuming tasks (which makes project management not scalable, especially for small teams), inaccurate results in testing phases, subjective evaluations, and authority constraints when phases are divided across teams. The work presented by Nogueria, Ana et al. [6] uses machine learning techniques to formulate strategies that lead to even higher product quality.

Although the adoption of DevOps has shown indisputable benefits, its implementation is challenging [7]. This challenge is significantly more complex when the implementation considers ongoing projects with developed practices and habits. The first issue faced is dealing with a currently established development routine [8], as having to re-establish current procedures and having the teams follow it can be met with resistance. The second issue is cost, as there are infrastructure considerations involving an initial cost in budget, skill development, and time taken for integration and maintenance of the tools required [9,10]. Finally, the third issue is the communication between the developer and operational teams, as continuous deployment can demand additional awareness of the different systems [11].

In recent years, the development of smart contracts has become popular thanks to the increasing interest in cryptocurrencies. Researchers have found interesting uses for smart contracts beyond the new currencies thanks to this revived interest. Smart contracts are protocols that enable and enforce contracts made between several parties on a blockchain. For this reason, the distributed fashion of blockchain creates particular requirements for elaborating the contracts [12,13]. As shown in [14], constant changes in the DevOps process can cause unexpected delays, which can be a big issue in the case of smart contracts, given that the language is changing fast [15] and with it, the need to implement new and more complete tools and development strategies. However, the viability of implementing a few steps of DevOps on an Ethereum blockchain has been proven by Wöhrer, and Zdun [16]. Implementing all the DevOps steps is important to guarantee a project’s success. Al-Mazrouai and Sudevan [17], Marchesi, Marchesi and Tonelli [18], Lenarduzzi, Lunesu and Tonelli [19] have all also suggested other processes using agile development as a base. These authors further validate that it is possible to establish a framework to work with blockchains and smart contracts.

In the end, however, implementing DevOps requires moving from old methods to a newer platform, which is even harder on smart contracts, for reasons previously stated. This paper proposes a structured process for implementing DevOps for smart contracts. In particular, the contributions of this paper are the following:

### 1.1. Contributions

There is an imperative need for a framework to guide the implementation of DevOps for smart contract development. This paper, therefore, deals with it by presenting:First, this paper presents a systematic analysis of applying a DevOps routine to smart contract development. In particular, we analyze how this practice may be adopted to continuously and securely integrate smart contract development into existing software. For this analysis, we go over the state-of-art of DevOps routine, its phases, and how to adapt them to use them in blockchain-based developments. We then survey software tools used in each DevOps Cycle’s phase, technologies for deploying blockchains and developing smart contracts, and known security vulnerabilities on smart contracts.As a result of the previous analysis, this paper presents a framework to implement DevOps to smart contracts and the interaction of its phases. To highlight its practicability, we analyze a set of tools for each stage. In concrete, we include a brief description of these tools and their potential use in implementing the proposed framework. Additionally, we present a case use to which the framework may be applied.

### 1.2. Paper Structure

The remainder of the article is organized as follows: Section 2 presents an overview of DevOps, and it also describes multiple software tools related to each phase in the DevOps cycle, while Section 3 describes multiple blockchain platforms and related tools. After these descriptions, Section 4 presents the security issues in developing smart contracts. Section 5 introduces our proposal for applying DevOps to smart contracts. Section 6 presents a summary of the framework using a use case to guide the steps required to run DevOps on smart contracts. Section 7 describes some implementation issues that require attention and consideration when implementing this framework. Finally, in Section 8, conclusions are drawn, and future research topics are listed.

## 2. Background

This section presents background material on DevOps.

### 2.1. DevOps

DevOps is a software development practice focused on improving the cooperation between the different teams working on a project, especially the development and operation team, from which the name came [2]. Modern development organizations require entire teams of DevOps to automate and reduce the gap between the development team and the operation team. At the same time, this process needs to acknowledge the interdependence of the different teams to produce faster results for software products and services in an organization [20]. DevOps can be defined in five major phases, as shown in Figure 1: Continuous Planning (CP), Continuous Integration (CI), Continuous Testing (CT), Continuous Deployment (CD), and Continuous Feedback and Monitoring [21,22].

CP is the phase that handles dynamic planning, where events are constantly assessed to determine the best action to take according to the specific event. This phase allows plans to be agile and adapt to moving conditions [22]. The process consists of a cycle where managers plan small steps, execute them and get feedback. These steps allow the team to react and adjust the current plan to match the feedback. As a result, the phase generates a prioritized product backlog, and participants can change this prioritization where adjustments can be made at any point [23]. CI allows the integration of multiple activities on a partial automation process going from source control to project building [24,25]. One of the CI process’s most significant advantages is that a clean code build is always the result. The result from the CI phase can vary from a compiled executable, an image, a container, or a library to be used on another project.

CT is the next phase of the DevOps process, where multiple tests are done over the source code and the compiled build during the CI phase. These practices have evolved from functional testing to include static code analysis and dynamic code analysis for security and performance issues. These analyses allow the detection of possible defects that developers must address as soon as possible. Systems’ users access the results of this activity in real-time without depending on each other [26], leading to more software features and finding bugs much faster. After the successful completion of the CT phase, the CD phase happens. The artifact’s deployment goes into the different environments for the product. The activities inside the CD phase do not handle the security of the generated artifacts, confidentiality, or infrastructure issues [24]. They do, however, facilitate updating a system component, allowing feedback from the different teams to be made available to the developer more frequently, given the automation of the process [27]. CD, however, can be divided into two types: Continuous Deployment and Continuous Delivery. The ending activity defines the difference. Continuous Delivery is not fully automated, as it finishes delivering an artifact capable of deployment. Continuous deployment, however, ends with the final artifact being pushed to a system without any decision-making.

Finally, CFM starts once the build has been deployed on a system. As Fayollas, Camille et al. [28] suggested, this could be included in the CD phase. However, to differentiate the most, a new phase is defined. Here, there exists a valuable chance to observe the behavior and usage of the system through different qualitative parameters to react to any bugs or improvements that can be done over the source at an early stage of the deployment [21].

There is, however, a significant point to consider when implementing the tools required by a DevOps process called security. As mentioned before, not only is it a part of the DevOps process to guarantee the security of the artifact being deployed, but the security of the application itself needs to be considered. As mentioned by Düllmann, Thomas et al. [29], security is also a concern in the tools themselves. There must be mechanisms to prevent identity spoofing, data tampering, repudiation, information disclosure, denial of service, and elevation of privilege.

### 2.2. DevOps Tools

This subsection describes software tools for each phase in the DevOps cycle. DevOps tools are the software in charge of helping with the different parts of software development, such as version control, building, testing, and deployment. Tools are tailored for specific activities of the process.

It is essential first to discuss Source Control Management (SCM) tools. Working on a shared project with a local directory is satisfactory at the start of the project. However, as it starts growing, i.e., other participants start joining and contributing to it, sharing the code changes becomes convoluted, and hence a way to do it is a need [30]. That does not imply that no benefits can be obtained from an SCM if working alone since a sole member may benefit if one wishes to recover a previous code version in case something goes wrong. An SCM tool stores the code in repositories, versions the code, and helps distribute the work between project members. Some SCM technologies are Git, Subversion (SVN), Concurrent Versions System (CVS), Vesta, and Mercurial.

The second set of tools handles the orchestration between the multiple services. These tools coordinate the processes required to compile a project. These ensure that the predefined steps are done for both success and error cases. Sometimes the orchestrator can be built-in in another tool, such as Gitlab (a Git repository manager) with its own CI/CD [11]. However, there are tools dedicated to this function, such as Jenkins, Tekton, Travis-CI, Circle-CI, and AppVeyor [14,16].

The third tool is not mandatory but is highly encouraged to have and handles the testing of the application. As mentioned before, testing should now include static code analysis and dynamic code analysis without forgetting about unit testing. The used code framework should cover unit testing. However, the other testing tools require other tools. A static code analysis tool is responsible for flagging preconfigured programming and style errors in the source code. Meanwhile, a dynamic code analysis tool flags potential vulnerabilities in a running artifact.

### 2.3. DevOps Cloud-Based Tools

Emerging cloud technologies influence the paradigm for applications, which has changed as the capability to develop and deploy on the cloud has increased the scalability and reliability of the systems [31]. Cloud providers have had to adapt to these changes causing the support of DevOps tools on the cloud to increase and allowing support for the DevOps process while minimizing the deployment downtime. The architecture that handles all of this is the microservice architecture [14]. One key factor that allows the implementation of the microservice architecture in the cloud is virtualization, which can be done through virtual machines and containerization. Some of the cloud providers that offer DevOps tools services are:**Microsoft Azure Pipelines**: Pipelines simplify hardware and VM management using Microsoft’s agents and allow users to automate code builds and deployments with Pipelines. Azure offers support for every major platform and tool through container jobs. Important to note is that Azure allows the user to do a deployment to multiple cloud providers and on-premise machines [32].**Google Cloud Build**: It is a fully managed platform to build, test, and deploy code from multiple source code repositories. Cloud Build permits the user to deploy the build to the platform of choice. Important to note is that Google offers “Binary Authorization” to perform deep security scans, enforce standardized container release practices and verify images to ensure no tampering is done on deployment [33].**Alibaba Cloud DevOps Pipeline (Flow)**: An automated delivery pipeline service that offers tools to go from continuous integration to continuous deployment. Flow benefits include integrating multiple cloud repositories, code and security scanning, and deployment to public clouds or self-hosted environments [34].**IBM Cloud Continuous Delivery (CCD)**: IBM CCD platform allows user to handle their pipeline entirely through tools integration. It offers issue tracking, source code repository, and web IDE in addition to the tools required to build, test and deploy. IBM CCD offers deployment exclusively to the IBM Cloud [35].**Amazon AWS CodePipeline**: It is a platform to build, test, and deploy code based on a defined release process model. It is based on three services: AWS CodeBuild (code building and testing), AWS CodeDeploy (deployment to AWS servers or on-premise servers), and AWS CodeStar (user interface for configuration) [36].**Redhat OpenShift Pipelines**: OpenShift Pipelines is a Kubernetes-native CI/CD solution based on Tekton. Therefore, this allows each pipeline step to run on its container. Openshift Pipelines can build, test, and deploy applications to public cloud platforms and on-premise [37].

Table 1 compares the analyzed cloud providers that a developer should consider when starting a smart contract development project.

## 3. Blockchains and Smart Contracts

As part of the framework proposal, this paper aims to contribute with a set of tools and backgrounds that they can select from for starting a smart contract development project. For a development project, readers can adequately select a set of blockchain and smart contracts that better suit the project’s goals.

Blockchains worldwide have become the base for digital currencies containing blocks protected from manipulation and alteration. Any block has information regarding the previous block and timestamp. Blockchains implementations are free from alteration as it is impossible to change the data in a block [38]. First-generation blockchains, like Bitcoin, introduced cryptocurrency transactions as their only function. Meanwhile, second-generation blockchains, like Ethereum, have introduced the possibility of building and deploying software executed by the members of the blockchain [39].

The programs on the blockchains are called smart contracts, which are the core of a blockchain service. A smart contract is, as previously stated, an immutable software code that runs on top of the blockchain. Currently, Ethereum is the most popular blockchain for smart contracts [15], but other blockchains with smart contract support are shown in recent studies [40].

Bugs and security concerns are issues that smart contracts need to solve to have a bigger adoption rate. Security concerns regarding smart contracts have appeared in recent years, as shown by different studies [41,42,43]. Bugs are an issue on both cost, as running smart contracts on a blockchain has a transaction cost [44], and security, as unsafe programming can allow an attacker to run undesired code [45]. Since smart contracts convert software programming instructions and due to requirements occurring in multiple scenarios, many smart contract platforms have become available to solve business requirements [42]. Some of these blockchains are [40]:**Ethereum**: One of the most popular open-source public blockchain platforms with a cryptocurrency called ETH or Ether. Ethereum is the oldest smart contract platform, allowing developers to build decentralized apps through its Ether. Ethereum Virtual Machine (EVM) software stores and executes all smart contracts, with Solidity as the relevant programming language. However, this blockchain suffers from concurrency issues which developers are working to reduce [46].**Hyperledger Fabric**: A private blockchain platform with smart contract features that first became available as an enterprise blockchain platform. It is an open-source Linux project which supports the collaborative development of blockchain-based distributed ledgers. This blockchain architecture bases itself on a microservice architecture for an appropriate deployment. Hyperledger developers have access to tools that allow them to develop smart contracts more efficiently and quickly [47].**New Economy Movement (NEM)**: A platform for private blockchains focused on building solutions for requirements using a cryptocurrency called XEM, which can be traded but not used as payment. Created initially in 2015 based on another blockchain called NXT [48,49]. Unlike other blockchains, NEM uses a proof-of-importance (POI) mechanism instead of a proof-of-work (POW) mechanism. NEM offers several advantages to the users, such as ease of deployment, deep customization, performance, and complete security.**Stellar**: It is a blockchain-based platform that facilitates economic transactions over boundaries. For the Stellar blockchain, smart contracts manifest as Stellar Smart Contracts (SSC) [50]. The crypto coin used in Stellar is Lumen [51]. A Stellar Smart Contract (SSC) is a composition of connected and executed transactions using various constraints with a processing time between three and five seconds. Stellar allows users to create, trade, and send digital representations of all forms of money (i.e., bitcoin, dollars, and pesos) while securing this with the Stellar Consensus Protocol (SCP).**EOS**: It is a blockchain-based platform designed to develop scalable and secure applications with smart contract capability [52]. EOS provides decentralized storage of enterprise solutions to solve the scalability issues faced by Bitcoin and Ethereum. A difference between the EOS platform and others is that it eliminates all users’ fees and uses a Proof-of-Stake (PoS) algorithm.**Corda**: An open-source blockchain project, designed for business from the start, allows users to transact directly with smart contracts [53]. This practice streamlines business processes to reduce transaction costs and record-keeping. The R3 Corda platform represents the smart contract corresponding to real-world contracts. It is an agile and flexible platform that can scale to meet business requirements. Applications built on Corda, CorDapps are designed and developed to transform businesses across various sectors, including insurance, healthcare, finance, and energy.

Table 2 has an abbreviated comparison of the blockchains discussed before.

## 4. Smart Contracts Security

In general, the vulnerabilities and security issues are essential for this framework’s design because smart contract developers need to know how to design the security checkpoints in the DevOps pipeline. Knowing them is critical for better outcomes at the end of the framework’s use.

In addition to the security risks posed against distributed ledgers, such as 51% attack [54] against blockchain based on mining [55,56], as well as emerging threats as cryptojacking [57], there are other risks directly against smart contracts. The distributed and immutable characteristics of a smart contract in a blockchain had consequences when faults in them caused economic impacts in multiple cases [45,58]. Such risks pressed developers to hasten the creation of multiple tools to mitigate these impacts. However, not all tools have been kept up to date or made accessible to a larger public, as shown by Lopez, Antonio et al. [59]. There is also the difference in programming language between the blockchains that only makes them available for some blockchains [40]. It is also worth mentioning the analysis of the tools developed are built for static code analysis instead of a dynamic code analysis [44].

There are doubts about the current tools available for smart contract programmers, which has caused proposals such as SolAnalyzer [60], SolAudit [58] and SuMo. However, the biggest concern is the need for articles concerning employing DevOps for a smart contract pipeline.

### 4.1. Smart Contract Vulnerabilities

Information on vulnerabilities of Solidity Smart Contracts has been collected in multiple studies such as Villar Lopez et al. [59], Badruddoja et al. l [43], Dika et al. [44], Akca et al. [58]. Most studies focus on and explain the most critical vulnerabilities presented in this section.

**Reentrancy**: This vulnerability is considered one of the most severe. The reentrancy vulnerability relies on the interaction between two smart contracts. If, through smart contract dependencies, a contract hands over the control to another contract, it allows this second contract to call back into the first contract before the first initiated interaction between them is completed. Therefore, the second contract could have an action to refund gas and do this operation multiple times to empty the balance of the smart contract of the target. The correct order of operations during a balance transfer is essential to prevent this attack.**tx.origin**: tx.origin is a global variable used in Solidity smart contracts containing the account’s address that sends the transaction. The vulnerability, named after this variable, aims to identify the user who initiated the chain of interactions between contracts. This action is done during the authorization method to spoof unauthorized users as authorized and obtain leveraged privileges in the contract.**Callstack depth exception**: Contracts have a call stack limit of 1024. Therefore it is possible to make a smart contract execution fail by making external calls and exceeding this maximum stack call size. An attacker can use this call as an advantage to produce an output that suits them the best during a contract execution if the call stack exception is not handled correctly by the contract.**Timestamp dependence**: A contract that depends on the timestamp can be vulnerable if used in a vital contract call. An attacker can manipulate the timestamp to produce an output that suits them best.**Transaction-ordering dependence**: This vulnerability is a prevalent security bug in the smart contract that consists of relying on the order of transaction execution. This vulnerability can have the attacker be the smart contract owner or the miner. It consists of making the gas price of the transaction change during the execution of the smart contract because the attacker sends a transaction that modifies the price before the transaction being executed finishes.**Gasless send**: This situation happens when, under certain conditions, the gas sent to execute a contract call is not enough to cover the call. This situation generates a “gas exhaustion” exception and, therefore, a transaction failure. The exception happens if the recipient’s contract has a fallback function with a large code base. It is crucial to throw an exception if a failure based on gas consumption happens and to ensure that the gas requirements for contracts’ calls are not too high.**Call to the unknown**: It happens when a Call, Send or DelegateCall primitive of the Solidity language is called inside a contract. However, this one internally uses a function defined inside all contracts (as part of the Solidity language) but is not found in the miner’s environment during execution.**Overflow and underflow**: This vulnerability can occur when transactions do not check the input data to verify it is an authorized input. Smart contract overflow occurs when the value provided exceeds the maximum value defined for its data type. In the case of Solidity, it is a 256-bit number. In the case of underflow, it is trying to achieve the same by using numbers under the limit permitted by Solidity.**Short address attack**: This vulnerability occurs in Ethereum Virtual Machine (EVM). It permits padded arguments that allow an attacker to send a crafted address that leads to a contract exploit. The EVM will add zeros to the end of the encoded arguments to make up for an expected length of 20 bytes (applied only when the argument is less than 20 bytes). This attack is not explicitly made on the Solidity contracts themselves but on the third-party applications that interact with them. It is usually an issue when third-party applications interacting with smart contracts do not validate the inputs.

### 4.2. Analysis Methods

For developing secure software, especially in the case of smart contracts, it is essential to make the code error-free. To achieve this level, developers need to involve a wide array of security testing [61]. Security testing for vulnerability detection methods divides into two significant testing methodologies: static vulnerability detection and dynamic vulnerability detection. These analyses can be specialized to follow specific standards according to the sector where the program’s execution will occur in [62].

#### 4.2.1. Static Vulnerability Detection

Static software analysis is a way of studying a program from its compiled binary code without executing it. The basic concept common to all static analysis tools is checking the source code to identify specific coding patterns that often lead to vulnerabilities [63]. Manually checking for these patterns is an option, but automated tools are more effective than manual options. There are several analyses done when doing static analysis [59].

**Control analysis**: Focuses on the flow of the different calls in a structure. The analysis revises the calls done in a process, function, method, or subroutine.**Data analysis**: Concerns using defined data to ensure data objects are correctly operating.**Fault/failure analysis**: Analyzes faults and failures in the data components used inside the code.**Interface analysis**: Verifies the interfaces between solutions and applications to determine that the components interact appropriately.**Pattern recognition analysis**: Searches for portions of code known to contain potentially vulnerable code.

#### 4.2.2. Dynamic Vulnerability Detection

Dynamic software analysis studies a program during service execution to find vulnerabilities that can only be detected upon the execution of a service [64]. This process consists of running multiple inputs in the program to crash it [65]. There are several analyses done when doing static analysis:**Fault localization**: Searches for faults in the code by giving multiple random values to tests to see if the tests fail with a specific value.**Memory errors and concurrency errors analysis**: Searches for resource and memory leaks during race conditions on multi-threaded programs.**Performance analysis**: Traces software applications at run-time and captures data to identify the causes of poor performance.

### 4.3. Blockchain Development Tools

A tool can be defined as a concept, technology, software system, template, framework, or library that aims to help design, development, and maintenance of a software product [66]. With this in mind, multiple tools used during the development of smart contracts are shown in Table 3 and Table 4, with information regarding IDEs and Security Tools.

Table 3 organizes the tools classified as IDE, which support software developers in their implementation work [82]. For these tools, depending on the blockchain, options for local development only or both local and web-based IDE were found. However, no blockchain offers a web-based-only approach. Table 4 presents the tools classified as per the kind of security analysis they may perform. Tools executed for static and dynamic analysis of the code created for smart contracts can be found in that table. Both tables include information regarding the available blockchain and the last time they were updated.

**Table 4 sensors-23-00541-t004:** Blockchain development tools for security analysis.

Blockchain	Tool Name	Classification	Description	Last Updated
Ethereum	Oyente [83]	Static Analysis	An Analysis Tool for Smart Contracts	Nov 2020
Ethereum	Solgraph [84]	Static Analysis	Visualize Solidity control flow for smart contract security analysis	Jan 2019
Ethereum	MadMax [85]	Dynamic Analysis	Ethereum Static Vulnerability Detector for Gas-Focussed Vulnerabilities	Jun 2021
Ethereum	Manticore [86]	Dynamic Analysis	Manticore is a symbolic execution tool for analysis of smart contracts and binaries.	Mar 2022
Ethereum	Mythril [87]	Dynamic Analysis	Security analysis tool for EVM bytecode.	Apr 2022
Ethereum	ContractLarva [88]	Dynamic Analysis	Runtime verification tool for Solidity smart contracts.	Mar 2022
Ethereum	SolMet [89]	Static Analysis	A static analysis tool for calculating OO-style source code metrics for Solidity smart contracts.	Nov 2020
Ethereum	Vandal [90]	Static Analysis	Static program analysis framework for Ethereum smart contract bytecode.	Jul 2020
Ethereum	Securify [91]	Static Analysis	A security scanner for Ethereum smart contracts.	Sep 2021
Ethereum	Slither [92]	Static Analysis	Static Analyzer for Solidity	Apr 2022
Ethereum	Ethlint [93]	Static Analysis	Code quality & Security Linter for Solidity	Sep 2019
Hyperledger Fabric	Revive-CC [94]	Static Analysis	Static analysis tool for Hyperledger Fabric smart contracts written in Go.	Jul 2020
Hyperledger Fabric	Blockchain Analyzer [95]	Data Analysis	Analyze ledger data stored within a Hyperledger Fabric peer.	Feb 2020
Hyperledger Fabric	Chaincode Analyzer [96]	Static Analysis	CLI tool to detect the codes which can be risks potentially such as nondeterminism in smart contract in Hyperledger Fabric.	Feb 2020

## 5. DevOps Framework for Smart Contracts Integration

This section presents the proposal for the application of DevOps to smart contracts. When used with both traditional software and blockchain smart contracts, the underlying DevOps concepts and techniques are strikingly similar. However, some phases are different while working with blockchains because of the limitations. In more detail, we outline the suggested DevOps phases as they apply to blockchains and smart contracts, along with the tasks involved in each phase and useful indicators that provide ongoing feedback. When it comes to tasks, we describe each one in full, including the input and output that are anticipated. In addition, important indicators that provide feedback for the different teams throughout the entire process will be taken into account and described in their own area.

### 5.1. Preparation

While the preparation itself is not a component of the DevOps approach, some of the actions conducted prior to beginning with DevOps are. Also, we consider this phase important to assure the quality and security of the following phases. It’s important to remember that a specialized team with the capabilities to take decisions should handle this step alone. At this beginning stage of the phase, a number of tasks should be finished, including:

#### 5.1.1. Analysis of Current Systems

The first phase should be an analysis of the existing resources, and the person or group in charge of it should provide all of the input. Repurposing the current resources requires a thorough analysis of the development team’s existing systems. This action can result in less training time because the team is already familiar with the present procedures, as well as a means to save money because no new system will be needed. The analysis of the current artifact and container repositories, the development tools (both for version control and automated testing), and the tools available for general work management are the results of this analysis, which is presented as a report to determine whether these tools will be compatible with the new blockchain project. A report containing no information would also be a valid result because it would indicate the need to buy or make plans not to use the unavailable tools (not recommended).

#### 5.1.2. Defining the Quality Strategy

As development consists of both functional and non-functional requirements, the recommendation is to work with a development and a quality environment before releasing the new smart contract onto the blockchain (if possible, as this depends on the blockchain). The input for these tasks consists of the identified quality metrics defined for the project (if they exist), with the output consisting of the testing strategy and the release strategy while ensuring the use of the distinct environments.

#### 5.1.3. Securing the Development Process with the Respective Tooling

With the input taken from the two previous tasks, this task validates the security practices required for the defined strategy. Security not only implies the integration of security tooling for the code but also for the tooling systems used by the team. If the team has no tools available, as an empty report is considered valid, the recommendation for the whole secured environment must be given (version control, artifact management tool, automated testing, and building).

#### 5.1.4. Validation of Existing Code Base and Dependencies

For this task, the information of previous source control and the repositories of existing systems are taken to validate the code base for compliance and inspect the dependencies for security flaws. The output of this step should be the code requiring manual validation and dependencies requiring updates due to security failures.

#### 5.1.5. Setup of Tools Environment

Considering all input taken from the previous steps, now a decision must be taken on the minimum necessities for the project and allow growth according to the given budget for tools that would allow saving time due to simplifying tasks. With all decisions made, this line of tasks ends up with an environment set up to start DevOps.

### 5.2. Continuous Planning

The DevOps first phase and the one that generates will generate the input for the following phases is the CP phase. As smart contracts should focus on a single task, developers should prepare to plan according to this ideology, where one prefers multiple small smart contracts [16]. This planning saves on the number of transactions done and thus reduces the execution costs.

#### 5.2.1. Task Discovery

The first task of the whole DevOps strategy will be to collect the information regarding the requirement needed to deploy. This result is considered a gap, and the multiple small tasks required to complete the gap are defined. This step is done by a single senior developer or a group of developers.

#### 5.2.2. Task Planification

The second task of the whole DevOps strategy will be management work. The project manager is informed of the tasks required to complete the expected functionality. These tasks must be registered in a work management flow tool to keep track of the activities in progress (e.g., Jira, Gitlab, ClickUp), which will serve as output for this task.

#### 5.2.3. Task Assignment

This final task will now take the activities registered on the previous task as input. Again, a senior developer or team will assign the responsibility based on the available human resources. They consider the task’s difficulty to assign to a developer according to their skills. This task then leaves an output of assigned activities with the requirement to be done by the developer.

#### 5.2.4. Indicators

These indicators will help provide a better understanding of the capacity of a team to fulfill the tasks created.

Resources Available: The total amount of resources assigned to a project. This doesn’t necessarily mean only human resources but other types of resources too (computational, tools, third-party supports).Assignment Time: The time taken between task creation and task assignment.Lead Time: Time taken between task assignment and task completion. This can only be calculated once the full development cycle is completed.Feature Prioritization: Reinforces the value of DevOps, where constant iterations are being done to meet user demands. This will determine if the tasks being assigned meet the needs of those features being utilized the most.

### 5.3. Continuous Integration

As explained previously, CI allows the integration of multiple activities on a partial automation process going from source control to project building. The input for this step is the activity list previously defined during the CP phase to solve the gap. The general flow, as previously mentioned, involves developing and building the code.

#### 5.3.1. Code

Developing new smart contracts involves taking the activities planned during the CP phase to solve the requirements needed. Various programming languages (e.g., Solidity, Java, C) can be used depending on the blockchain. Depending on the blockchain, the IDE developers will work with also varies (e.g., Remix, VSCode, HyperledgerComposer). The output of this step will be the code that solves the requirement given.

#### 5.3.2. Unit Test Creation

After being done with the code, or in parallel, the unit tests should get coded too. This step will guarantee that the code is ready for production and behaving as the developer expected. The input for this task is the current developed code, and the output is the unit tests created to assert that the code is behaving as was initially expected.

#### 5.3.3. Commit

Different Git repositories (e.g., Github, Gitlab, Bitbucket) and SVN repositories (e.g., SourceForge, CloudForge) can store the developed code. The code, the input for this case that is published, will then be submitted for a merge request. This process involves giving human approval to take the submitted changes and apply them to the main code. This step works as a peer review to at least get a second pair of eyes to validate the changes. No automatic acceptance is recommended here as the chance for a peer review before testing is lost. In the end, the step output will be the merge requested code.

#### 5.3.4. Build

The build step includes all the steps required to generate the artifacts needed for execution from the source code. The compiler and instructions to execute the source code change depend on the blockchain and programming language used. The input taken for this step is the merged code that is now in the code repository. An established CI software will then do the build (e.g., Jenkins CI, Travis CI, Gitlab CI/CD), which will involve automating the actions required to generate the final output of the CI phase: the compiled smart contract.

#### 5.3.5. Indicators

These indicators will help provide a better understanding of task fulfillment by developers.

Unplanned Work: Represents on-demand changes done while work is in progress due to unidentified situations during task discovery. This could also be due to unexpected situations occurring due to ongoing changes. No matter the reason, however, this is a metric that should be accounted for. Aimed at being a low value, as it represents proper discovery work being done and no changes being accepted once work has started.Change Volume: Determines the extent to which code was changed when fulfilling a given task. Ideally, the change volume should remain low implying only the task objectives are being changed.Test Coverage: Constitutes the percent of functionality in a smart contract that is covered by unit tests. A low value implies the unit test creation step is being skipped, which then will lead to unexpected bugs.

### 5.4. Continuous Testing

The CT phase has steps that can be executed only after the CI output. However, some of the analyses done during this phase can be executed even during the CI phase. To separate responsibilities between phases, it is assumed that all testing will get done after the CI phase. In this phase, a series of automated tests review the output of the smart contract. As in traditional software, multiple layers are available for testing in smart contracts. Layers can dictate a subdivision, i.e., the contract, the data, or the blockchain’s consensus [97]. When a test involves functional testing, there should be a simulated local temporary blockchain where the smart contract can be deployed and tested.

#### 5.4.1. Static Analysis

This step involves taking the source code as input and using a subset of tools to test it for different issues (e.g., checking for errors, checking structural problems) without executing the smart contract. For example, Splinter is one tool that can be run against the source code to test the smart contract. Other examples of tools are in Table 4. The test metric should be as error-free as possible, and the output of this test is either a pass or no pass.

#### 5.4.2. Dynamic Analysis

Different tools are available to work on dynamic analysis. Some take the source code and work as a static analysis tool, while others take the generated output and work on this. These tools examine the code for potential bugs, undesirable patterns, and patterns that could lead to possible errors during the smart contract execution. Some of these tools are Mythril, Manticore, Vertigo, and Echidna. This process naturally comes with the possibility of false positives, so a minimum required metric for the output of pass or no-pass should be planned.

#### 5.4.3. Private Blockchain Deployment

Before unit testing or integration testing can happen, there is a need for deployment on a blockchain. The suggested approach due to cost and control is to have a private blockchain already created which has been done in the setup of tools environment mentioned in the Preparation phase. This task will take as input the build previously generated and deploy it into the test blockchain where the following steps can connect. The output of this task will be a success or failure of the deployment.

#### 5.4.4. Unit Testing

The third type of testing to be executed is unit tests. According to the survey by Chakraborty et al. [98], this step should not be skipped as it was the testing methodology that most commonly found issues on blockchain software. The ideal scenario for unit tests includes: all methods covered, all inputs validated, transactions reverts checked, and access privilege verified [16]. This task takes as input the unit tests previously developed. The output for this step includes a metric informing the tests passed. The advantage of unit testing is the complete automation and parallel testing execution using frameworks such as OpenZeppelin or Truffle.

#### 5.4.5. Integration Testing

This type of testing validates the interaction of various components [99]. In the context of smart contracts, this involves setting up multiple smart contracts in specific states to validate the proper behavior of the smart contract. This activity can be time-consuming, mainly due to the manual setup needed for each case. The input for this activity consists of the smart contract functionality to test, and the output consists of a metric determining the tests passed.

#### 5.4.6. Indicators

These indicators give insight into the testing being executed over smart contracts. Low or high values, depending on the indicator, can reflect future issues that will be faced if deployment is done instead of revisiting the scenarios.

Test Pass Rate: If code releases consistently fail unit tests, this suggests that teams are ignoring safe practices, and work must be done to correct those issues. This value should always remain high, implying functionality works as expected according to the tests created.Error Detection Rate: Represents the errors and warnings detected by the static and dynamic analyses. The detection rate value aim is to be as low as possible, implying that the common errors found in code are accounted for.Escape Rate: Represents the value of changes creating changes in functionality that were not caught in testing. A high value implies that the testing must be reviewed to determine where the issue lies.

### 5.5. Continuous Deployment

As described by Górski, the CD aims to enable on-demand software release [100]. However, the smart contracts’ complexity and interaction with the blockchain strain the phase. In this phase, the input obtained from the CT phase will be the metrics that will allow the process to decide if the compiled smart contract will be released and deployed.

#### 5.5.1. Release

The release step is a decision step, where all the inputs from the testing get evaluated against previously decided acceptance metrics. If the step decision is “pass”, the smart contract is considered releasable. This decision is followed by generating all the files needed to release the smart contract in a blockchain. The output for this step will be the files prepared for deployment.

#### 5.5.2. Deploy

The deployment step can be manual or automatic, working on a pull (deploy when you need) or push (deploy when you release) configuration. The release decision on how to operate the deployment depends solely on the operation team, as the smart contract would then be immutable in the blockchain [101]. The input for this step is the released contract generated on the Release step, and the output is whether the smart contract has been successfully deployed.

#### 5.5.3. Indicators

These indicators provide information regarding deployment and the team’s ability to respond to scenarios regarding this.

Deployment Time: The time between release and deployment. Deployments can occur with high frequency if tasks are simplified enough, but these times should remain relatively constant. Any dramatic increases in deployment time will warrant further investigation,Recovery Time: The time indicating the team’s ability to respond appropriately to issues during deployment. It would mean little if issues are found promptly but not followed by an equally rapid recovery time.Failed Deployment Rate: This value represents the ratio of deployment failures and is related to the previous indicator. The value should be kept as low as possible to indicate that a low amount of issues are being generated.

### 5.6. Continuous Feedback and Monitoring

Assuring smart contracts are secure before deployment is preferable when possible [102]. This step in the DevOps process verifies proper behavior and generates feedback for the CP phase. This phase still has work to be done in the current state as the available tools need to permit extensive monitoring. The input for this phase is the deployment notification from the deployment step, which will allow the monitoring to start and generate feedback.

#### 5.6.1. Monitoring Smart Contract

Monitoring smart contracts, as previously stated, is a challenging task as it depends on the capabilities of each blockchain to give the information required to allow monitoring. Assuming data is available, possible metrics expected are transactions versus actual transactions, the overall cost per transaction, and transaction speed. Also, even when blockchains can resist data loss or unintended data manipulation to a certain extent, there is a need to monitor the data integrity. This monitoring could be done automatically by generating automated reports with the compiled data.

#### 5.6.2. Monitoring Permissions

In cases where authorizing accessing data is done, via methods, features, or permissions, an attacker or a misconfiguration can lead to a breach of privacy. Therefore, since it is required to ensure that confidential data remains as so, the team must be able to check on data access. It is possible to monitor data access manually or automatically by using state comparison tools to raise warnings in case of differences.

#### 5.6.3. Monitoring Client

Having access to a client node for the smart contract allows the monitors to validate the proper behavior of the smart contract on that specific client node. Multiple validations on distinct nodes allow the generation of reports. However, this implies the work has to be done manually and, therefore, will have a high resource cost.

#### 5.6.4. Feedback

This step involves taking the information from the monitoring steps and generating a feedback report. This report will then be sent to a team lead to evaluate utilizing the same work management flow tool defined in the CP phase as a new task for the team lead.

#### 5.6.5. Indicators

Not the most important, but highly valuable information is contained in these indicators. This determines whether a good job was executed by the different teams and allows for feedback to be given appropriately.

Compliance: This highlights the difference between the deployed work and the planned work. A high compliance value ensures that expectations are being met.Ticket Volume: This concept reflects alerts (a monitoring user generates) to indicate bugs or unexpected functionality. An increased ticket volume suggests issues in the deployed code and not caught in testing.Performance: A key indicator for any application. Code deployed should not affect the performance of previous functionality and should not generate unaccounted slow performance times.

## 6. General Use Case

This section describes a framework summary using a use case to guide the steps required to run DevOps on smart contracts. The guidance image is in Figure 2. This figure explains the overall process and steps to take in each.

The next is the continuous planning phase, which consists of two tasks: planning and assignment. Issue tracker software covers these functionalities. Examples of software dedicated to issuing tracking are Atlassian Jira, HubSpot, ClickUp, and Backlog. However, other applications like Gitlab or Microsoft Teams have issue-tracking capabilities.The next phase is the continuous integration phase, which consists of coding tasks. For the smart contract coding, it is necessary to create unit tests for the smart contract and commit to the repository hosting the smart contract or groups of smart contracts. Once completed, a build process using an automation server should verify that the recent commit will allow the code to build successfully. The repository, with technologies mentioned in Section 4.3, can be hosted by software such as Github, Gitlab, VisualSVN Server, or Apache Subversion, among others. Meanwhile, the automation server, as mentioned in Section 4.3, can be done by software such as Jenkins, Tekton, Travis-CI, Circle-CI, and AppVeyor.The next phase is continuous testing, which tests the code directly using static analysis tools. Dynamic analysis tools are also run on the compiled code. Some tools are referenced in Table 4. The automation server should do these tests once the build successfully generates a report on possible bugs. Finally, manual and automatic integration tests are available using blockchain simulators such as Ganache or Geth once all tool testing finishes.The release of the smart contract is generated, which involves the creation of the needed files to publish the smart contract on a blockchain. After this, the files are deployed onto the blockchain. Everything in the continuous deployment phase should be left to the automation server, or at least as much as possible.The continuous monitoring of the released smart contract starts operating. The proposal is to do this by monitoring statistics related to the smart contract and the data generated by it and its users. Also, monitoring related permissions to the smart contract must be constant, so there is a verification regarding undesired permission changes. A feedback report is filed from the operations team to send back to the development team.

## 7. Implementation Issues

This section describes some implementation issues that require attention and consideration when migrating from traditional software to a smart contract-based application, which would lead to the application of this framework.

### 7.1. Secure Deployment and Integration

This segment describes implementation issues to consider to make a secure deployment and integration of smart contracts.

Define, design, and implement a mechanism to allow existing software to request/execute a smart contract method. Given that a complete overhaul of the software architecture is expensive and not accounting for the cases where specific technologies do not allow this to be done, the goal would be to reutilize as much as possible from the current software. However, a mechanism that would connect current software with a smart contract with minimal change is out of the scope of the proposed framework.Define, design, and implement a mechanism to authenticate and authorize requests to access resources in the smart contract from existing software. Such a mechanism should correspond to identities created in the existing software to accounts from the blockchain. This part would help extend the authorization model to control what identities may or may not access resources deployed in implementing or updating smart contracts. Authentications or authorizations should not be lost under any circumstance for a system. For all migration cases, the framework’s deployment step should consider any migration on the authentication and authorization model to guarantee that data is kept.Define, design, and implement a mechanism to properly handle/integrate the different versions of a smart contract deployed into the blockchain. Since a contract version is unavailable for modification after deployment, such a mechanism should help direct existing software requests to the smart contract’s correct version. This mechanism would act as a proxy, which mediates requests from existing software to the smart contract. During the deployment of the updated smart contract, the deployment of this proxy should also be updated. Whether this mechanism is another smart contract or another tool is left to the implementer. However, it is recommended to find an automated way according to the implementation.

### 7.2. Secure Monitoring

This segment describes implementation issues regarding the monitoring of smart contracts, as the transactions occurring on execution time increase the need for proper monitoring and analysis. Sometimes the information required to be monitored in a smart contract is unavailable depending on the implementation. One way to accomplish monitoring is to look at all contract transactions. However, that may need to be sufficient, as message calls between contracts are not recorded in the blockchain. A monitoring mechanism should:Define, design, and implement metrics to measure events related to the operation of a smart contract. An event is a convenient tool given by smart contracts to record executions in the contract. Events that were emitted stay in the blockchain along with the other contract data and are available for future audits. A mechanism should be available to constantly seek this data to transform it into a visual form. This mechanism would work as feedback for the different teams working on implementation.Define, design, and implement a mechanism to utilize the data from the smart-contract operation. Such a mechanism should access the data extracted mentioned in the previous point. This data can then be given as machine models, fed and deployed to measure the previously defined metrics. The objective of this mechanism is to make this data available in visual form for the different teams involved in the implementation.

## 8. Conclusions and Future Works

DevOps is a proven development strategy applied to traditional software development that shows high potential for the blockchain smart contracts development cycle. As shown in our framework, applying these same principles is considered with extra precautions to handle the nature of smart contracts. The strategy is created with overall security while associating tools with specific activities to make it easier to identify the step. It also has the benefit of contributing to both the repeatability and portability of the framework. It is considered a capability desired by many organizations for security and business purposes.

Further work is required to demonstrate the mentioned repeatability and portability using more tools to prove these characteristics. However, there is a confidence that the changes required for the application in multiple tools will be exclusively on the tool implementation rather than the steps required. With the security goal of both the framework and the smart contracts, we expect the integrated vulnerability checks to create a much higher awareness of security issues during development. This goal will force developers to fix vulnerabilities as soon as possible.

As blockchain technology continues to move forward, so will the requirement to build software around them. However, testing the released smart contracts cannot be reduced as they are considered a critical practice that cannot and should be considered. In this regard, after stating the best practices for smart contract development and the overall steps, this research has found some critical steps that require further research. This future research should evaluate two main points: first, the strategy to identify the points used to evaluate the output of the further analysis, depending on the blockchain, to specify whether the smart contract steps of the CT phase can be successful. Second, to automate the monitoring of the smart contracts as the current proposal is a task to be done manually. Ultimately, the teams integrating DevOps into their practice will face challenges to ensure reliable and secure blockchain smart contracts, to which this paper places its contribution.

## Figures and Tables

**Figure 1 sensors-23-00541-f001:**
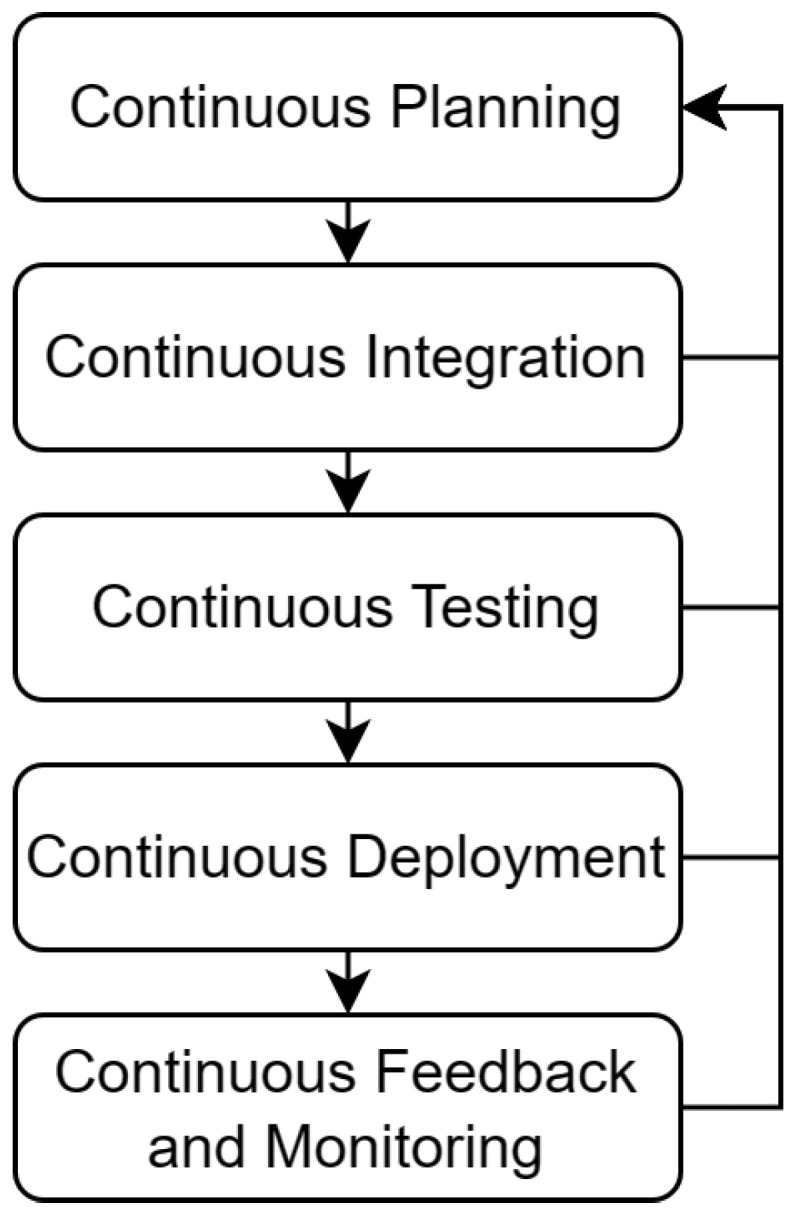
DevOps phases.

**Figure 2 sensors-23-00541-f002:**
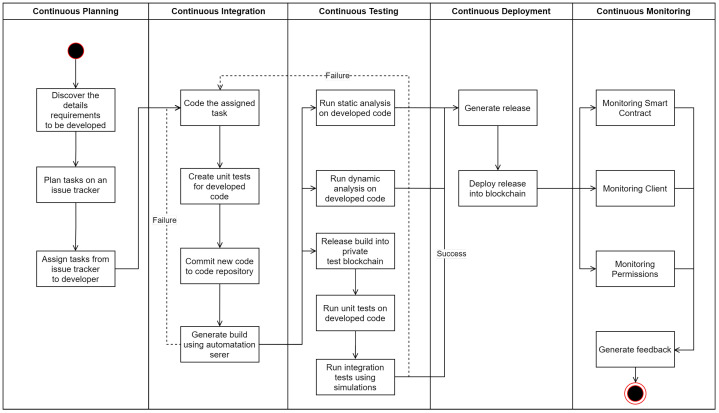
Framework example.

**Table 1 sensors-23-00541-t001:** Comparison of cloud providers.

Cloud Provider	On-Premise Deployment	Multi-Cloud Deployment	Additionals
Microsoft	Yes	Yes	Extension marketplace
Google	No	Yes	Binary Authorization
Alibaba	Yes	Yes	
IBM	No	No	Issue tracking, Web IDE
Amazon	Yes	No	
Redhat	Yes	Yes	

**Table 2 sensors-23-00541-t002:** Comparison of most used blockchains.

Blockchain	Open Source	Supports Cryptocurrency	Miner Participation	Smart Contract Language	Consensus Mechanism	Scalable
Ethereum	Yes	Yes	Public	Solidity	Proof of Work	No
Hyperledger Fabric	Yes	No	Public, Private	Java, Solidity, Golang	Proof of Work	Yes
NEM	No	Yes	Private	Java	Proof of Importance	Yes
Stellar	Yes	Yes	Public	Solidity, Javascript, Java, Go	Stellar Consensus Protocol	Yes
EOS	Yes	Yes	Public	C++	Proof of Stake	Yes
Corda	Yes	No	Private	DAML, Kotlin, Java	Validity and Uniqueness	Yes

**Table 3 sensors-23-00541-t003:** Blockchain development tools for IDE purposes.

Description	Tool Name	Classification	Blockchain	Last Updated
Web-based IDE with built-in static analysis, and a test blockchain virtual machine	Remix [67]	Web-IDE	Ethereum	Apr 2022
Web-based IDE that lets you write, compile, and debug your smart contract, powered by Loom Network	EthFiddle [68,69]	Web-IDE	Ethereum	Jun 2021
A Cloud-Based Multi-Chain IDE that provides debugging, testing and deployment one-stop services, developers don’t need to install extra tools while working on smart contracts	ChainIDE [70]	Web-IDE	Ethereum	Mar 2022
A customizable development environment for Ethereum with hot reloading, error checking, and first-class testnet support	Replit [71]	Web-IDE	Ethereum	Apr 2022
Visual Studio Code is a lightweight but powerful source code editor which runs on your desktop and is available for Windows, macOS and Linux.	Visual Studio Code [72]	Local-IDE	Ethereum, Hyperledger Fabric, Corda	Apr 2022
IntelliJ IDEA is an integrated development environment written in Java for developing computer software.	JetBrains IDEs [73]	Local-IDE	Ethereum, Corda	Apr 2022
Remix Desktop is an Electron version of Remix IDE. It works on Linux, Windows, & Macs.	Remix Desktop [74]	Local-IDE	Ethereum	Dec 2021
Truffle is a development environment, testing framework and asset pipeline for multiple blockchains.	Truffle [75]	Local-IDE	Ethereum, Corda	Apr 2022
An application development framework which simplifies and expedites the creation of Hyperledger fabric blockchain applications.	Hyperledger Composer [76]	Local-IDE	Hyperledger Fabric	Aug 2019
Stellar uses no specific IDE to work with. You install the stellar-sdk for the language you want to program with which means you can interact with the network through the API	Stellar-SDK [77]	Local-IDE	Stellar	Apr 2022
A web IDE integrated with various tools required for EOSIO in a unified graphical application.	EOS Studio [78]	Web-IDE	EOS	Feb 2020
Desktop version for EOS studio for Windows, Linux and Mac.	EOS Studio Desktop [79]	Local-IDE	EOS	Sep 2019
IDE providing developers with a personal single-node EOSIO blockchain for development and testing.	EOSIO Quickstart Web IDE [80]	Web-IDE	EOS	Jun 2020
A source code editor for Windows, Linux and macOS.	Zeus IDE [81]	Local-IDE	EOS	May 2021

## Data Availability

Not applicable.

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
