# Peer review of "Continuous and Secure Integration Framework for Smart Contracts"

_sensors, 2023, doi:10.3390/s23010541_

Round 1

Reviewer 1 Report

The research topic is definitely very current.

There are few issues to be addressed.

Use Case in section 7. should present real example of using proposed framework in building a smart contract. Maybe renaming section – "Possible Use Case" or something similar.

Also Figure 2 is barely visible. Landscape view would provide better visibility.

 This sentence in the Introduction is unclear:

"Al-Mazrouai has also suggested other processes using agile development as a base, and Sudevan , Marchesi, Marchesi and Tonelli, Lenarduzzi, Lunesu and Tonelli."

Author Response

English was reviewed across the document to fix grammar and spelling mistakes. Also, thanks to these comments and the comments from the other reviewer, we were able to fix several sections of the paper to explain better the methodology, the results and to justify the use of some of the sections of the paper.  Specific responses to the reviewer's comments are the following:  

  1. Use Case in section 7. should present real example of using proposed framework in building a smart contract. Maybe renaming section – "Possible Use Case" or something similar. Response: the section was renamed as recommended.
  2. Also Figure 2 is barely visible. Landscape view would provide better visibility. Response: we fixed the quality of the picture. 
  3. This sentence in the Introduction is unclear: "Al-Mazrouai has also suggested other processes using agile development as a base, and Sudevan , Marchesi, Marchesi and Tonelli, Lenarduzzi, Lunesu and Tonelli." Response: we fixed the sentence. 

Reviewer 2 Report

It is interesting to review this work for CICD of smart contracts, particularly with the security concerns. However, it has several concerns:

1. The contributions list is weak. Please elaborate the technical details for the state-of-the-art, particularly focusing on the continuous and secure integration part.

2. Software tools discussion in section 3 is not critical and has failed to demonstrate the landscape of such leverage. Please clearly identify the relationship and connection with the topic, with/without their practical usage, pros and cons.

3. Section 4 and 5 appear to be a literature review. I would expect a better discussion for their technical merit towards the goal of this work.

4. Not sure about the logic of Section 6, since it has been quite superficial. Meanwhile, how does it connect with section 7? I would appreciate a good discussion of the use case.

5. Section 8 should be better improved with insights, a further discussion will be appreciated.

Author Response

We fixed some English grammar and spelling mistakes across the document. We also improved the connections between the sections and the paper's overall organisation. 

  1. The contributions list is weak. Please elaborate on the technical details for the state-of-the-art, particularly focusing on the continuous and secure integration part. Response: we extended the explanations of the contributions. 
  2. Software tools discussion in section 3 is not critical and has failed to demonstrate the landscape of such leverage. Please clearly identify the relationship and connection with the topic, with/without their practical usage, pros and cons. Response: the section was integrated into Section 2 as it could be considered mostly background content. We still highlight the importance of this section as the paper is also considered useful for researchers to pick the appropriate administrative tool. 
  3. Section 4 and 5 appear to be a literature review. I would expect a better discussion for their technical merit towards the goal of this work. Response: We added some introductions to these sections to improve their connections to the rest of the paper
  4. Not sure about the logic of Section 6, since it has been quite superficial. Meanwhile, how does it connect with section 7? I would appreciate a good discussion of the use case. Response: we added more information to Section 6 to connect it better with number 7.  We added new extra details to give more depth.
  5. Section 8 should be improved with insights, a further discussion will be appreciated. Response: we added extra discussion for each point to add more depth to the section. 

Round 2

Reviewer 2 Report

nill.